# Autoimmune Hepatitis and Fibrosis

**DOI:** 10.3390/jcm12051979

**Published:** 2023-03-02

**Authors:** Rinaldo Pellicano, Arianna Ferro, Francesca Cicerchia, Simone Mattivi, Sharmila Fagoonee, Marilena Durazzo

**Affiliations:** 1Unit of Gastroenterology, Città della Salute e della Scienza Hospital, C.so Bramante 88, 10126 Turin, Italy; 2Department of Medical Sciences, University of Turin, C.so A.M. Dogliotti 14, 10126 Turin, Italy; 3Institute for Biostructure and Bioimaging, National Research Council, Molecular Biotechnology Centre, 10126 Turin, Italy

**Keywords:** autoimmune hepatitis, liver fibrosis, liver diseases

## Abstract

Autoimmune hepatitis (AIH) is a chronic immune-inflammatory disease of the liver, generally considered a rare condition. The clinical manifestation is extremely varied and can range from paucisymptomatic forms to severe hepatitis. Chronic liver damage causes activation of hepatic and inflammatory cells leading to inflammation and oxidative stress through the production of mediators. This results in increased collagen production and extracellular matrix deposition leading to fibrosis and even cirrhosis. The gold standard for the diagnosis of fibrosis is liver biopsy; however, there are serum biomarkers, scoring systems, and radiological methods useful for diagnosis and staging. The goal of AIH treatment is to suppress fibrotic and inflammatory activities in the liver to prevent disease progression and achieve complete remission. Therapy involves the use of classic steroidal anti-inflammatory drugs and immunosuppressants, but in recent years scientific research has focused on several new alternative drugs for AIH that will be discussed in the review.

## 1. Introduction

Autoimmune hepatitis (AIH) is a chronic immune-inflammatory liver disease. AIH has a universal distribution, but its prevalence varies according to sex, age and ethnicity [1]. Although it can affect individuals of both sex and age, it is most frequent in women with a bimodal age-related incidence curve (a first peak in children and adolescents, and a second one in middle age) [1,2].

AIH is usually considered a rare disorder. However, its prevalence is rising worldwide, probably due to a change in lifestyle, but also to improved diagnostic skills that have reduced the number of undiagnosed AIH [2]. Based on recent epidemiological data, it is estimated to range between 160–170 cases/1000.000 inhabitants in Europe and 200 cases/1000.000 in North America [1,3].

AIH causes are not fully understood. However, current knowledge points to the development of the disease in genetically predisposed individuals (strong association with genes encoding the HLA class II DRB1 alleles), after their exposure to triggering factors (e.g., infections). In this case, the autoimmune injury occurs through molecular mimicry mechanisms and is favored by the decreased T-cells regulation [1].

Clinical manifestation is extremely varied and can span from mild forms with unspecific symptoms, such as malaise, lethargy, nausea, anorexia, pruritis and jaundice, to severe or even fulminant hepatitis [3]. Approximately 12–35% of patients are totally asymptomatic while one-third of them already have cirrhosis at diagnosis, regardless of the presence of symptoms [1].

AIH is characterized by interface hepatitis on liver histology, hyper-gammaglobulinemia, elevated aminotransferases and circulating autoantibodies [4]. Based on autoantibody profile, AIH can be classified in three subtypes: AIH type 1 (AIH-1) represents the most frequent (about 80%) and is associated with detection of antinuclear antibodies (ANA), smooth muscle autoantibodies (SMA) and perinuclear anti-neutrophil cytoplasmic antibodies (p-ANCA). AIH type 2 is quite rare (3–4%, mainly children) and is characterized by the presence of anti-liver/kidney microsomal antibody type 1 autoantibodies (anti-LKM1), anti-LKM type 3 (anti-LKM3) and/or antibodies against liver cytosol type 1 antigen (anti-LC1). Lastly, AIH type 3 (10–20%) is associated with detection of antibodies against soluble liver antigen (SLA) or liver-pancreas antigen (LP) [1,4].

Autoimmune response together with other inflammatory factors are responsible for the progression of liver damage, which can lead to fibrosis and even cirrhosis. Fibrosis is related to both duration and severity of the inflammatory activity [5]. Thus, the adequate assessment of this condition, to improve AIH prognosis, represents a challenge and increasing knowledge regarding the underlying fibrotic mechanisms is essential to develop new anti-fibrotic agents.

In this review, we discuss recent non-invasive methods for fibrosis detection and provide an update on the therapeutic strategies which are potentially useful in reversing the fibrotic condition in AIH patients.

## 2. Fibrogenesis in AIH

Fibrosis can be defined as the consequence of the reparative response occurring in liver tissues after an injury, which leads to an excessive accumulation of scar tissue (extracellular matrix, ECM) in the hepatic parenchyma [5].

In conditions of prolonged liver injury such as AIH, fibrogenesis dominates over fibrolysis, resulting in deposition of excess ECM, mainly collagens. This is accompanied by a downregulation of metalloproteinase (MMP) secretion and activity, and by an increase in tissue inhibitors of MMPs (TIMPs). These alterations promote tissue damage and drive a secondary fibrogenic response, such as allowing hepatic stellate cells (HSCs) and myofibroblasts to migrate and proliferate. Ultimately, changes in immune cell composition and angiogenesis lead to severe architectural modifications that compromise organ functions [6].

Several factors are implicated in hepatic fibrosis pathogenesis and progression. These include: inflammatory activity, cells apoptosis and oxidative stress (Figure 1).

In AIH, the inflammatory state is primarily instilled by autoimmune activity on liver tissues. However, the hepatocytes’ response to inflammation also plays a decisive role in the pathophysiology of fibrosis. In fact, liver damage involves the recruitment of both pro- and anti-inflammatory cells, such as monocytes and macrophages mediated by the activation of Toll-like receptors (TLRs) expressed in HSCs and Kupffer cells. Inflammatory cells amplify the response through the production of cytokines and chemokines, which increase HSCs stimulation. TLR itself promotes the transforming growth factor beta (TGF-β) activity, facilitates its pro-fibrotic actions on HSCs and myofibroblasts, and increases the production of Tumor Necrosis Factor alpha (TNF-α), Interleukin-1 beta (IL-1β) and chemokine ligands [7]. Fibrogenic cytokines (e.g., TGF-β, IL-1β, IL-17, IL-20) and chemokines (e.g., CCR2, CCR5) facilitate fibrosis development by stimulating trans-differentiation of HSCs into myofibroblasts, which are the main source of ECM production [8,9,10]. Moreover, TGF-β can expand the extracellular matrix content by repressing natural killer (NK) cell activity, thus resulting in decreased HSCs apoptosis and fibrolytic MMP inhibition [10].

Generation of apoptotic bodies represents another key factor implicated in fibrotic progression in AIH. Apoptosis is the principal mechanism of cell death in liver parenchyma, mainly actuated through the extrinsic (receptor-mediated) apoptotic pathway. However, in case of oxidative stress, the intrinsic (mitochondrial-mediated) pathway can also occur because of its activation by reactive oxygen species (ROS) [7].

Oxidative stress is the condition whereby ROS production exceeds their elimination capacity. It is implicated in many chronic liver diseases, playing an important role in promoting liver fibrosis in AIH, too [4,7,10]. In inflammatory liver conditions, ROSs are generated by myofibroblasts and Kupffer cells when they incorporate apoptotic bodies. ROS in turn stimulates the intrinsic apoptotic pathway and activates HSCs implicated in fibrogenesis [7].

Pemberton et al. observed that individuals with AIH-1 had elevated levels of oxidative agents compared with healthy subjects [4]. They also detected a positive correlation between ROS levels and total necroinflammation, interface hepatitis and fibrogenesis, leading the latter to be considered as a marker of liver damage. In addition, AIH subjects seemed to have an altered antioxidant status, with reduced levels of many antioxidant compounds such as glutathione, selenium, vitamin A, vitamin D and vitamin E [4].

## 3. Fibrosis Assessment

The assessment of liver fibrosis can be detected using several methods, which are summarized in Table 1. Although all these procedures are of great interest in the research field, currently only liver biopsy and elastography are recommended in clinical practice [11].

### 3.1. Liver Biopsy

According to the European Association for the Study of the Liver (EASL) Guidelines, liver biopsy is the “gold standard” for the assessment of liver fibrosis and it allows the evaluation of its localization and quantification [11]. Although some histologic patterns are characteristic of AIH, such as interface hepatitis, emperipolesis (intrusion of one intact lymphocyte into a hepatocyte) and hepatic rosette formation, there is no specific histologic finding of AIH [1,2,12]. Other frequent histologic patterns in AIH are Kupffer cell hyaline granules, prominence of plasma cells in portal tracts, and the relative predominance of plasma cells over lymphocytic inflammation [2].

Panlobular hepatitis, massive and bridging necrosis are usually observed only in acute and severe forms [13]. In biopsies performed upon acute onset, the characteristic histological pattern is panacinar hepatitis, which resembles those of drug-induced hepatitis [14]. Alternatively, pericentral necrosis may be found, resembling acute toxic injuries [15]. These lesions have been proposed by the US NIH Acute Liver Failure Study Group for diagnostic criteria for AIH presented as acute liver failure [16].

Liver biopsy is an important element for the diagnosis of AIH, as it offers the possibility to diagnose AIH also in patients without circulating autoantibodies. Moreover, histologic analysis permits the assessment of different fibrosis stages, which can be a useful information both in treatment decisions and in follow-up strategies [13]. 

However, liver biopsy is quite an invasive and expensive technique. Therefore, clinical attention has progressively moved towards more non-invasive and cheaper methods, able to replace liver biopsy in assessing hepatic fibrosis [17].

### 3.2. Non-Invasive Laboratory Tests

Many non-invasive laboratory tests are used to estimate liver fibrosis in chronic viral hepatitis and non-alcoholic fatty liver disease. However, most of these non-invasive laboratory tests have not yet been used or well validated in the field of AIH [7].

#### 3.2.1. Serum Biomarkers

Fibrosis biomarkers can be considered as indirect indexes of the fibrotic process. Although they are less precise than liver biopsy, they can be very useful in clinical practice for therapy guidance. Fibrosis biomarkers include various cytokines, chemokines and enzymes, but at present, evidence is available for only two compounds in AIH diagnosis: TGF-β and angiotensin-converting enzyme (ACE).

TGF-β is one of the major cytokines, which is overexpressed in AIH patients and directly correlates with inflammatory and fibrotic activities [18,19].

The role of renin-angiotensin system in fibrosis development is supported by higher serum levels of ACE among AIH subjects [20]. In fact, 56 U/L and 64 U/L of ACE concentrations are considered accurate cutoff values for severe fibrosis (respectively stage B2 and B3), while 84 U/L is used as a cirrhosis cutoff [21].

Promising results were also obtained regarding Lysil Oxidase Like 2 (LOXL2). Recent studies observed higher serum concentrations of this enzyme, which promotes crosslinking of collagen and elastin, in patients with Primary Sclerosing Cholangitis (PSC). Moreover, LOXL2 levels seemed effective in detecting advanced fibrosis and cirrhosis in PSC subjects [22]. Studies are required to investigate whether elevated LOXL2 hepatic expression, as well as circulating levels, correlate with fibrogenesis in AIH, too.

Lastly, recent preclinical studies suggested the possible role of some miRNAs as biomarkers of liver fibrosis. Serum levels of miR-133a were significantly elevated in AIH fibrotic mice, while levels of miR-122 and miR-21 were negatively correlated with liver fibrosis [23]. To this regard, circulating extracellular vesicles, which are an important source of nucleic acids and proteins released by the damaged liver, can develop into promising non-invasive biomarkers for AIH [24].

#### 3.2.2. Scoring Systems

Scoring systems for fibrosis assessment are usually calculated through the combination of two or more laboratory tests. Most of the used scores include: FibroTest, Aspartate Amino-Transferase (AST)/Platelet Ratio Index (APRI), Fibrosis-4 Index (FIB-4) and red cell distribution width to Platelet Ratio (RPR).

FibroTest combines five serum biochemical values (α2-macro-globulin, haptoglobin, apolipoprotein A1, γ-glutamyl-transpeptidase and total bilirubin), which are adjusted for subject age and gender. It is widely used as a diagnostic score in Hepatitis B Virus (HBV) and Hepatitis C Virus (HCV) patients, because of its good sensitivity and specificity in assessing advance fibrosis (70% and 93%, respectively) [25,26]. It has also showed an acceptable performance (AUC > 0.80) in cirrhosis detection among NAFLD patients [27]. However, the use of FibroTest in autoimmune liver diseases is still uncertain.

APRI is another simple index that is calculated with two routinely available laboratory tests: AST and Platelet (PLT) concentrations [21]. It is widely used for fibrosis assessment in case of viral hepatitis, especially HBV (AUC = 0.74 for advanced fibrosis) and HCV (AUC = 0.91 for cirrhosis) [28,29]. APRI also showed good accuracy in advanced fibrosis detection among NAFLD patients (AUC = 0.66–0.76) [30,31]. Nevertheless, data concerning APRI use in AIH are not favorable as many studies showed poor diagnostic accuracy of the score in detecting advance fibrosis in autoimmune patients [32,33], although it seems to be correlated to the liver fibrosis stage [34].

Similar considerations apply to the Fibrosis-4 Index. FIB-4 is based on four factors: age, aspartate aminotransferase (AST), alanine aminotransferase (ALT) and platelet. It is calculated using a relatively simple formula: (age [years] × AST [U/L])/(platelet [10^9^/L])(ALT [U/L])1/2) [35]. FIB-4 has been shown to be quite accurate in staging liver fibrosis in patients with viral hepatitis (AUC = 0.77–0.82) and NAFLD (AUC = 0.72–0.82) [28,31]. Instead, the accuracy of FIB-4 in AIH is more controversial as it has shown mediocre sensitivity and specificity [33].

Red cell distribution width to Platelet Ratio (RPR) is a novel index which considers two laboratory tests: red blood cell distribution width and PLT count. The former reflects variability rate of erythrocyte size and it is already considered a prognostic marker of various medical conditions [36]. The latter has long been known to be decreased in liver diseases, because of PLT sequestration [37], and it is included in various fibrosis scoring systems, such as APRI and FIB-4 [21,35]. Thus, the combination of these two values has been supposed to correlate with the fibrosis stage. RPR showed good diagnostic performance in assessing significant and advanced fibrosis in the case of viral hepatitis [38,39]. Studies concerning AIH are still scarce. However, preliminary data by Li et al. have indicated RPR as an accurate score for advance fibrosis assessment (AUC = 0.82) and as an independent risk factor to predict advanced liver fibrosis in AIH patients (OR 95% CI = 2.65 (1.383–5.170)) [40]. Of course, further studies are needed to better evaluate the diagnostic role of RPR and whether it can predict treatment response and long-term outcomes of AIH.

Therefore, though available scoring systems seem to be promising, few studies and their small sample sizes do not permit conclusions to be drawn about their utility in patients with AIH. Hence, EASL does not recommend the use of scoring systems in AIH clinical practice [17]. Moreover, it is important to underline that the diagnostic scoring system should be used to support the clinician in difficult cases and should not replace liver biopsy, which remains the gold standard for the assessment of fibrosis.

### 3.3. Radiological Procedures

Rather than laboratory tests, radiological techniques are widely used in ordinary practice. They are able to assess liver fibrosis based on ultrasound, magnetic resonance imaging and elastography. No conventional characteristics for AIH-related liver fibrosis have been described, as AIH imaging (described below) share features with other chronic liver diseases [41]. Moreover, radiological findings in AIH have never been reported systematically.

#### 3.3.1. Conventional Ultrasound

US allows visualization of liver parenchyma through sonographic features, and to monitor fibrosis progression as well. In fact, US permits detection not only characteristic changes in liver surface, such as nodularity and hepatomegaly, but it can also assess portal vein patency, diameter, flow speed and direction, besides ascites and splenomegaly [42].

Among AIH patients, US is the fundamental principle of Hepatocellular Carcinoma (HCC) surveillance. As 1–9% of cirrhotic patients develop HCC, both American and European Liver Associations recommend a semiannual ultrasound surveillance in these subjects [43]. Ultrasound is well accepted for this indication as it is a non-invasive, radiation-free and cost-effective technique with high sensitivity and specificity (40–80% and 80–100% respectively) [41].

However, conventional ultrasound is not adequate to detect early or moderately advanced fibrosis. For this reason, other US techniques such as elastography are commonly used in the diagnostic pathway.

#### 3.3.2. Elastography

Elastography is a non-invasive method in which US is employed to estimate liver stiffness (LF). The technique is based on the principle that US waves cross hard tissues faster than soft ones. Therefore, measuring shear wave speed allows assessment of liver fibrosis and even cirrhosis in patients with chronic liver disease [28,44,45]. Ultrasound elastography includes three main procedures: point shear wave elastography (pSWE), two-dimensional shear wave elastography (2D-SWE) and transient elastography (TE) [46].

In the former, SWE techniques compression and US waves are produced within hepatic tissues, while in TE a mechanical external impulse is used to generate them. Shear wave speed in a single point of the liver is measured using pSWE and it acquires a grayscale images of the organ. Indeed, 2D-SWE allows a wider hepatic area to be surveyed, bringing the operator more accurate results with anatomic B-mode ultrasound images and elastographic color maps. In both methods, liver stiffness measures are expressed in kilopascals (kPa). In patients affected by HBV, HCV and NAFLD, pSWE and 2D-SWE has shown high sensitivity and specificity for each stage of liver fibrosis assessment, although 2D-SWE had higher sensitivity for significant and advanced fibrosis [44,45,47].

Studies analysing diagnostic performance of pSWE and 2D-SWE in AIH are still scarce, but preliminary results seem promising. A study conducted by Park et al. on 49 AIH patients showed moderate sensitivity and specificity of pSWE for significant fibrosis (93.6% and 44.4%, respectively) and cirrhosis (63.6% and 86.8%) [48]. While analysis on a larger sample of AIH patients by Xing et al. observed that 2D-SWE had good performance in the assessment of liver fibrosis. In fact, liver stiffness had a strong correlation with the histological fibrosis stage (r = 0.71, *p* < 0.0001) with AUC 0.84, 0.84, and 0.94 for detection of significant fibrosis, severe fibrosis and cirrhosis, respectively. Calculated cut-off liver stiffness values for predicting significant fibrosis, severe fibrosis and cirrhosis were 10.0, 15.8 and 19.3 kPa, respectively. Moreover, 2D-SWE diagnostic performance was higher than that of other noninvasive serum biomarkers (APRI and FIB-4) [49]. Although necro-inflammatory activity may represent a confounding factor in predicting liver fibrosis, in this study it has no effect on liver stiffness measures of severe fibrosis and cirrhosis. In contrast, another pediatric study carried out by Galina et al. observed a strong association between liver stiffness and necro-inflammatory activity (*p* < 0.0001) in 33 pediatric AIH patients [50]. Thus, the results are still contradictory and more studies are needed to better understand this matter.

TE is the elastographic method most used in clinical practice. It is a novel technology based on ultrasound and low frequency elastic waves for the assessment of liver stiffness measurement (LSM) [51]. TE performance has been widely proven for many chronic liver diseases. In viral hepatitis, TE showed good diagnostic accuracy among HBV patients, with AUC = 0.82 for significant fibrosis and AUC = 0.91 for cirrhosis assessment [52]. Similar outcomes have been reported for HCV, where TE also showed a valid diagnostic accuracy for liver fibrosis staging [53,54]. Use of TE is also functional among NAFLD patients because of its good performance in both fibrosis and steatosis detection (AUC = 0.85 for advanced fibrosis and AUC = 79 for severe steatosis) [55,56]. Moreover, the large meta-analysis by Shi et al. proposed TE measurement as a predictor of chronic liver disease complications. In particular, TE had a valid diagnostic prediction for significant portal hypertension, with 90% sensitivity, 79% specificity and LS cut-off values in the range 13.6–34.9 kPa. Whereas performance for oesophageal varices prediction resulted in less precision (AUC = 0.84) [57].

Because of its widespread use in clinical practice, studies considering TE in AIH are more numerous. In a recent systematic review, Wu et al. showed that TE performed better than other non-invasive techniques, such as APRI and FIB-4, in detecting liver fibrosis among AIH patients. Moreover, TE had a good performance for fibrosis staging, with AUC = 0.90, 0.91 and 0.89 for significant fibrosis, advanced fibrosis and cirrhosis, respectively [32]. Another survey carried out by Guo et al. observed that staging cut-off values in AIH patients were not the same as those of other liver diseases. In fact, stage F2 had LSM cut-off of 6.27 kPa, F3 of 8.18 kPa and F4 of 12.67 kPa [58].

The timing for performing TE may also influence its accuracy. Indeed, TE had a better diagnostic performance in AIH patients who have been on long-term therapy than those who received a shorter treatment or none [59]. This fact can be due to higher levels of liver inflammation during the first months of disease, which are known to impact on stiffness measures [59]. Usually, inflammatory activity with high levels of aminotransferases can influence liver stiffness values, which can overestimate fibrosis readings in TE: The higher the grade of inflammation, the greater the influence on LS values [60]. Therefore, this problem is more notable in case of acute or fulminant viral hepatitis and during the early phase of any chronic liver disease. Although in available studies, it is rarely reported that hepatic inflammatory activity had a significant effect on LSM determination, Janik et al. tried to resolve this bias. According to their preliminary results, the combination of liver and spleen elastography may help in improving the diagnostic performance by reducing the inflammation bias [61].

Diagnostic accuracy of TE can also be compromised by being viscerally overweight and obese [46]. In order to avoid such difficulties, elastography can be performed using an extra-large (XL) probe. An XL probe allows improved reliability and validity of LSM among obese subjects, with overlapping results of normal weight ones [62]. Similarly, food ingestion prior to TE may alter measurement readings, increasing liver stiffness regardless of fibrosis state. Thus, it is recommended TE is performed at least 120 min after the last meal [63].

In conclusion, TE has been demonstrated to be a clever non-invasive method for fibrosis assessment in AIH. It is reproducible, rapid and easy to perform, minimally invasive and well tolerated by patients. Moreover, cost-effectiveness analysis has indicated TE as a valid alternative to liver biopsy, which allows a saving of up to $14.000/year for patients [64]. For all these reasons, EASL recommends the use of TE (together with transaminases and IgG) to monitor AIH course and stage liver fibrosis in patients on at least 6 months of therapy [17].

#### 3.3.3. Magnetic Resonance Imaging

Magnetic resonance imaging (MRI) is a non-invasive imaging technology able to produce 3D anatomical images through magnetism principles. MRI sequentially employs powerful magnets and radiofrequency currents to stimulate protons present in the human body, while some sensors detect the energy released by protons themselves. As MRI scanners do not require ionizing radiation, it is considered quite a safe procedure and it is largely used for the detection of various diseases and for treatment monitoring [65].

In the hepatological field, the most commonly used MRI-based techniques are magnetic resonance elastography (MRE) and magnetic resonance spectroscopy (MRS).

MRE combines magnetic resonance technology with low-frequency vibrations in order to create an elastogram showing liver tissues stiffness. Several studies have confirmed its good diagnostic accuracy for fibrosis assessment among NAFLD patients (AUC = 0.91 for advanced fibrosis), which was even better than that of TE [44,66]. Similar data were obtained for viral hepatitis, with MRE AUC = 0.94 and 0.97 for advanced fibrosis and cirrhosis, respectively [28]. The good performance of MRE in other chronic liver diseases has stimulated its evaluation for AIH, too. Available data are based on small surveys, but preliminary results seem promising. In fact, Wang et al. observed that MRE had superior accuracy compared to other non-invasive laboratory tests in advanced fibrosis and cirrhosis detection, with good sensitivity and specificity (89–92% and 96–100%, respectively) [67]. Moreover, according to a recent systematic review, MRE accuracy is not influenced by liver inflammation, either in treated or untreated patients [32]. Further studies on larger cohorts are required to confirm these data. MRE could thus develop into a new alternative for liver fibrosis assessment and staging in AIH.

MRS is a spectroscopic technique which allows observation of local magnetic fields around atoms. As liver cell membranes are composed of phospholipids, the analysis of phosphorus (31P) magnetic fields can be used to study hepatocytes state. The 31P-MRS have been experienced in various liver diseases, such as HCV and NAFLD, where it results have shown promise in assessing both fibrosis and inflammation states [68,69]. Recently, Puustinen et al. tested this technique among AIH patients [70]. Despite the small sample size (only 12 subjects), they observed that several phosphorus metabolites (e.g., phosphoenolpuryvate, phosphoethanolamine/phosphocholine ratio) correlated with inflammation grade, hepatitis activity and immunoglobulins levels, engendering a differentiation between patients with and without active inflammation possible. Moreover, 31P-MRS was able to detect advanced fibrosis and to diversify fibrosis stages, which can be useful for disease monitoring.

In conclusion, recent studies showed that both MRE and MRS are promising non- invasive methods for fibrosis assessment in AIH. However, further investigations are needed to evaluate their diagnostic accuracy in wider populations and to estimate the cost of their enhanced use. In fact, MRI-based techniques presently have limited availability in the clinical setting, because of their high cost and need for highly qualified staff.

#### 3.3.4. Computed Tomography

Computed tomography (CT) is a procedure that combines a series of X-ray images taken from different body angles in order to create a cross-sectional image (slice). Therefore, abdominal TC scan allows visualization of internal organs and their morphological findings.

In the hepatological field, although CT has shown effectiveness in assessing advanced liver fibrosis states, it was less accurate in cases of mild fibrosis [71,72]. For this reason, CT is mainly used among cirrhotic patients, where it allows precise detection of signs of portal hypertension, enlargement of portal vein, presence of collateral circles and esophageal varices [73]. CT has also shown good diagnostic accuracy in HCC detection. However, it is not recommended for routine surveillance as compared to US as it is more expensive and less safe (because of ionizing radiation exposure) [74]. Nevertheless, specific studies on AIH patients are still lacking.

## 4. Fibrosis Treatment

The goal of AIH treatment is to suppress fibrotic and inflammatory liver activities to impede the progression of the disease and to achieve a complete remission [75]. Therapeutic pathway can be divided into two phases (induction and maintenance), while remission is defined by the absence of clinical symptoms and the achievement of normal transaminases level [76]. The main diagnostic-therapeutic algorithm of AIH is summarized in Figure 2.

### 4.1. First-Line Therapies

Recommended first-line therapy for AIH patients includes the combination of two principles: an anti-inflammatory (prednisolone) and an immunosuppressive (azathioprine—AZA) [11]. American Guidelines recommend starting with administration of both molecules, while European ones suggest introducing AZA 2 weeks after corticosteroids [11,77]. After 4 weeks of induction therapy, a maintenance phase can be achieved with a fixed dose of prednisolone (10 mg/day) and AZA (10 mg/day) until transaminases level normalization [77]. Treatment should be continued for at least 3 years and for at least 24 months after biochemical remission. Histological resolution is normally achieved after reaching the biochemical endpoint [78].

However, relapses are quite frequent, and they usually occur in the first 12 months after stopping the treatment. Relapse is defined by the elevation of ALT > 3 times ULN and it does not require a histological confirmation. In order to re-induce remission, the standard AIH therapy (prednisolone + AZA) is also used among relapsed patients [11,79]. Not all patients respond to standard treatment and side effects can be seen in those who respond to the therapy.

Steroids induce a great deal of adverse effects, which range from cosmetic changes to diabetes, psychosis, hypertension, and osteoporosis. Although the association with AZA seems to lower the occurrence of side effects [80], EASL Guidelines suggest replacing prednisolone with budesonide in patients with present or expected adverse events.

EALS guidelines suggest replacing prednisolone with budesonide at a starting dose of 9 mg/die in patients with expected side effects. Manns et al. found a reduction in side effects caused by steroid use in the group treated with budesonide (26.0% vs. 51.5%) [81]. Furthermore, budesonide induced a much higher remission rate in this cohort of patients compared to prednisolone [81], even if it was ineffective in the presence of cirrhosis [82]. Thus, in prednisolone responders with side effects, a switch to budesonide can be considered; otherwise, it can be possible to administer high doses of AZA (2 mg/kg).

AZA intolerance is rarer, and it is frequently associated with cirrhosis. Side effects include arthralgias, fever, skin rash and pancreatitis. In case of AZA intolerance, the substitution with Mycophenolate mofetil (MMF) is recommended [11,83].

In contrast, non-responders are characterized by a reduction of transaminases below 25% after 2 weeks treatment. Non-responsiveness can vary from incomplete to null response. Fortunately, null-responders (with or without immediate severity) are quite rare. In the case of liver failure, prognosis is poor and overall mortality ranges from 19% to 45%; otherwise, increased dosages of standard therapy or alternative immunosuppression may be used [11]. Instead, patients with clinical, biochemical and histological improvements but no complete resolution (abnormal liver enzymes or interface hepatitis on a liver biopsy) are defined as incomplete responders. In this case, if the increased dosage of prednisolone and AZA does not prove effective, a second line therapy should be started.

### 4.2. Second-Line Therapies

Recommended second-line drug for AIH patients includes mycofenolate mofetil (MMF) [84]. MMF is a non-competitive inhibitor of inosine monophosphate dehydrogenase, an important rate-limiting enzyme in purine synthesis, and it has anti-T and -B cell effects [85]. Guidelines recommend 1 g of MMF twice daily, gradually increased from a starting dose of 500 mg twice daily to improve drug tolerance [86]. In the case of AZA intolerance, MMF in combination with prednisolone induces an effective response (rate from 43% to 88%) and rare side effects [83,87]. When reported, adverse effects include gastrointestinal symptoms, impaired wound healing and teratogenicity (in fact its use is strictly contraindicated in pregnancy) [88].

A recent meta-analysis shows that MMF treatment is effective and safe even in naïve AIH patients, with higher remission rates of aminotransferase and IGG levels and lower non-response rates than standard therapy [89]. However, the cost of MMF still limits the use of the drug [88].

### 4.3. Third-Line Therapies

If the second-line treatment is ineffective, difficult to treat patients can resort to third-line drugs, which include: calcineurin inhibitors (Cyclosporina A, Tacrolimus), mTOR inhibitors (Everolimus) and monoclonal antibodies (Rituximab, Infliximab) [84]. Calcineurin inhibitors (CIN) act by suppressing T reg activation and IL-2 production. The most used CIN are Cyclosporine A and Tacrolimus, which seem to be effective in refractory AIH patients [90].

Cyclosporine treatment (2–3 mg/kg/day) showed a high biochemical response rate (>80%). However, it has been tested in only a few studies, with limited sample size and brief trial duration [91,92]. Side effects include nephrotoxicity, neurotoxicity and higher risk of infection. Some authors have even suggested that Cyclosporine A might promote autoimmunity and may have a more immunosuppressive than anti-inflammatory capacity [93,94]. For these reasons, Cyclosporine A is recommended as an alternative therapy when there is no response to AZA and steroids [95].

More data are available for Tacrolimus. Several studies have shown that Tacrolimus therapy (0.1 mg/kg/twice day) could induce biochemical and histologic improvement in patients with steroid refractory AIH [96]. Although its effect is more powerful than Cyclosporina, it is often not sufficient to achieve full remission [84]. However, its fair efficacy and reduced nephrotoxicity make Tacrolimus a valid alternative to standard AIH therapy.

mTOR is an intracellular kinase involved in cell proliferation, motility and survival. Data concerning mTOR inhibitors (such as Everolimus) are scarce and limited to case series [84]. Their role in AIH must yet be explored but preliminary outcomes are promising for refractory disease [97].

Third-line therapies also include two monoclonal antibodies: Rituximab and Infliximab. They have been recently used in patients with refractory or difficult to treat AIH [98]. Rituximab induces depletion of B cells, which are involved in antigen presentation and T cell suppression [99] while Infliximab acts against the proinflammatory cytokine tumor necrosis factor alpha (TNF-α). Side effects of these monoclonal antibodies include hepatotoxicity and other immune-mediated disorders [100]. Thus, further studies are needed to establish safety profiles, dosage guidelines and monitoring strategies. For the moment, their use is recommended only in specialized centers with considerable experience in AIH therapy and monitoring.

### 4.4. New Therapeutic Perspectives

Recently, scientific research has focused on novel alternative medication for AIH. Some of these are described below.

#### 4.4.1. Pharmacological Agents in Development

Several novel drugs for AIH have been developed, but their efficacy is still under investigation.

JBK-122: This molecule has anti-inflammatory properties as it is an antagonist of toll-like receptor 4 (TLR4) involved in the production of proinflammatory cytokines. JBK-122 seems to be effective both alone and in combination with steroids. It is currently being employed in a phase 2 study in adults which are intolerant, refractory or ineligible for standard therapy [101,102]. The primary outcome of the study is to assess changes in ALT after 24 weeks of treatment.

VAY736: It is a human IgG1 monoclonal antibody. Its target is the B cell activating factor (BAFF), which is a regulator of B cell maturation and survival [103]. As previous investigations observed high BAFF levels in patients with AIH, some trials are now ongoing to assess the effect of VAY736 as an innovative therapy [102,104].

Interleukin-2 (IL-2): Former studies showed that IL-2 deficiency can lead to increased susceptibility to Treg apoptosis. Thus, exogenous administration (TRANSREG) may protect Treg from this process [105]. It is currently employed in a phase 2 clinical study that assesses the safety and efficacy in 12 autoimmune diseases, including AIH [103]. The primary outcome of the study is to evaluate the change in percentage of Treg on the eighth day compared to baseline.

Adoptive Treg cell transfer (ACT): Treg cells are involved in the control of immune response through several mechanisms [106]. However, quantitative and qualitative Treg deficiencies have been found in AIH patients. ACT is an innovative therapy in which autologous immune cells are modified before being reintroduced in the patient. A phase 1 clinical trial in children and adult with moderate disease and on standard therapy is ongoing [102]. The primary outcome of the study is biochemical and immunological remission.

#### 4.4.2. Antioxidants

As the goal of AIH treatment is to reduce liver fibrosis, many new therapeutics act by suppressing inflammatory activities.

Oxidative stress is higher among AIH patients than in the general population, and it promotes the deposition of ECM. However, antioxidants compounds are known to reduce ROS production, thus leading to a decrease in cell apoptosis and inflammation [4].

Vitamin D is widely known for its anti-inflammatory and antioxidant properties. Recent rat models showed that it has also a role in liver fibrogenesis. In fact, vitamin D inhibits HSCs proliferation and their conversion into myofibroblasts. It also suppresses the expression of genes that promote collagen production, hence reducing ECM expansion [107,108]. Therefore, it is not surprising that vitamin D deficiency has been associated with advanced fibrosis and severe interface hepatitis [109].

Epidemiological investigations have revealed that serum 25-hydroxyvitamin D concentration was frequently low among AIH patients [109]. This fact was primarily due to the impaired hydroxylation of vitamin intermediate metabolites by the damaged liver, but also other factors (e.g., reduced sunlight exposure, advanced age) may contribute.

This substantial scientific background seemed to support the use of vitamin D supplementation in AIH patients. However, clinical trials investigating the effect of vitamin D among AIH patient are scarce and a recent systematic review stated there is not enough evidence to determine whether its supplementation has any beneficial, harmful or neutral effect on chronic liver diseases [110].

Besides vitamin D, many other antioxidant compounds have been investigated. Among them, zinc supplementation showed promising results. In fact, in a recent study conducted by Moriya et al. on 38 AIH patients, zinc supplementation induced the improvement of serum fibrotic biomarkers (MMPs, TIMs) [111]. However, these data are in contrast with those of a larger meta-analysis, where the authors concluded that zinc did not provide benefits in patients affected by chronic liver diseases [112]. Although nearly one third of patients take a multivitamin supplementation after AIH diagnosis, there is still no evidence to recommend it [113].

#### 4.4.3. Gut Microbiota Modification

Recently, scientific attention has focused on intestinal microbiota in order to evaluate its relationship with AIH and therefore its possible role in the disease treatment. The contribution of gut microbiota in AIH onset and progression was confirmed by the finding of dysbiosis among AIH patients. In fact, compared to the general population, the microbiota of these subjects was characterized by low biodiversity and decreased anaerobes and Bifidobacterium species concentration [114,115].

A few preclinical studies tested the effect of eubiosis restoration on liver injury and inflammation with promising results. Both the use of probiotics and short-chain fatty acids seemed to facilitate AIH remission in rat models, but these data need to be confirmed in further clinical trials [116,117,118].

#### 4.4.4. Epigenetic Regulation

The search for new therapeutic strategies for AIH has also considered the epigenetic remodeling occurring in this disease. As diverse miRNAs are involved in the inflammatory pathways, the modulation of their expression may be effective in leading to a reduction in liver fibrosis, too. Several preclinical trials have recently confirmed this hypothesis. Ke et al. observed that the inhibition of miR-375 decreased apoptosis in Kupffer liver cells, while Wang and colleagues noted that downregulating miR-138 improved immune status in AIH mice [119,120]. The decreased apoptosis of Kupffer cells and the improvement of the immune status can lead to a reduction in liver fibrosis.

Instead, miR-143 is notably involved in inflammation and fibrosis by regulating the TGF-β-activated kinase 1 (TAK1) phosphorylation. Thus, TAK1 may become a good target for future genetic therapies against fibrogenesis [121]. Obviously, clinical application of possible TAK1 inhibitors requires further studies and, especially, the presence of both miRNA-143 and TAK1 in humans needs to be investigated. Epigenetic regulation of liver fibrosis in the AIH context through miRNAs, as well as other epigenetic actors such as DNA methylation, histone modification, noncoding RNAs (ncRNAs) and N6-methyladenosine (m6A) modification, seems a promising innovative strategy [122].

## 5. Conclusions and Future Perspectives

The knowledge of the pathobiology of liver fibrosis has significantly advanced during recent years. However, further investigations are still necessary to develop new technology for both early fibrosis assessment and treatment. Key challenges regard determining the pivotal molecular changes or patterns associated with AIH, understanding the complex interaction among genes and gene networks involved in promoting the disease and fibrotic outcome, finding biomarkers for early disease detection in order to develop, in a timely way, therapeutic interventions that restore homeostatic balance without significant side effects.

## Figures and Tables

**Figure 1 jcm-12-01979-f001:**
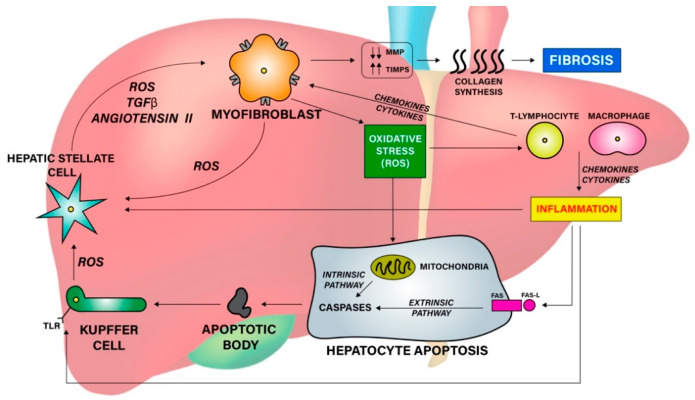
Liver fibrosis pathogenesis and progression. Chronic liver damage results in the activation of liver and inflammatory cells that lead to inflammation and oxidative stress through the production of mediators. This results in increased collagen production and extracellular matrix deposition leading to fibrosis. ROS: Reactive Oxygen Species. TGF-β: Transforming Growth Factor Beta. MMP: Metalloproteinase. TIMPs: tissue inhibitors of MMPs. TLR: Toll-like receptors. FAS-L: FAS ligand.

**Figure 2 jcm-12-01979-f002:**
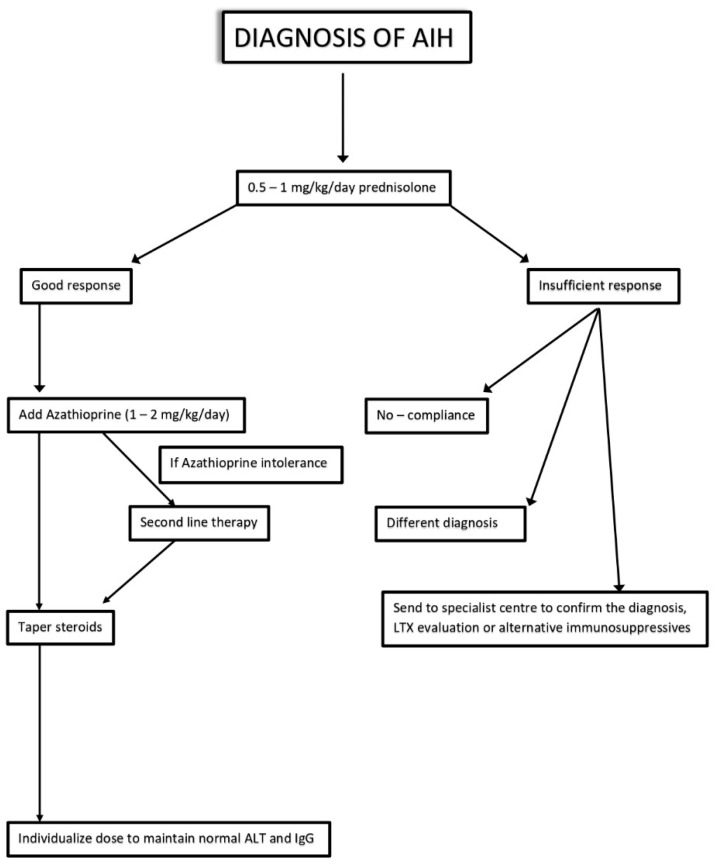
Diagnostic-therapeutic algorithm of AIH. ALT: Alanine Transferase. IgG: immunoglobuline G. LTX: liver transplant [77].

**Table 1 jcm-12-01979-t001:** Methods of diagnosis and staging of liver fibrosis.

General Method	Specific Procedures
Liver biopsy	-
Serum biomarkers	TGF-β Angiotensin-converting enzyme (ACE)
Scoring systems	FibroTest Aspartate Amino-Transferase/platelet Ratio Index (APRI) Fibrosis-4 Index (FIB-4) Red cell distribution width to Plateled Ratio (RPR)
Radiological procedures	Conventional ultrasound Elastography Magnetic resonance elastography (MRE) Magnetic resonance spectroscopy (MRS) Computed tomography (CT)

## Data Availability

Not applicable.

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
