# Peer review of "Autoimmune Hepatitis and Fibrosis"

_jcm, 2023, doi:10.3390/jcm12051979_

Round 1
Reviewer 1 Report
The review on autoimmune hepatitis and fibrosis by Pellicano et al is well-written. However, the following suggestions can further enhance the quality of the paper:
#1. Introduction The introduction could benefit from a discussion of the pathophysiology of autoimmune hepatitis, with a focus on genetic susceptibility and the absence of immunoregulatory function.
#2 The author should also emphasize that the diagnostic scoring system should only be used to support the clinical judgment in challenging cases and should not replace liver biopsies. Furthermore, the author should distinguish between autoimmune hepatitis and checkpoint inhibitor-induced hepatitis, including differences in histology and treatment approaches.
#3 Finally, the reference sources for the increase in prednisolone to 100 milligrams per day should be specified - this is not recognised in major guidelines for treatment of AIH (Figure 2)
#4 In the treatment section, the author should discuss the role of budesonide as an alternative to prednisolone in treating autoimmune hepatitis.
Author Response
We thank the Reviewer for the comments and suggestions, which give us the opportunity to improve our manuscript. Please find attached our point-by-point responses.

Reviewer 2 Report
The manuscript by Pellicano and colleagues nicely summarize our current knowledge on autoimmune hepatitis, a chronic immune-inflammatory disease of the liver, as well as the current methods of diagnosis and treatments for this rare condition. It is well-written, clear, and include a number of recent papers in the field. The paper covers the topic from a general point of view, and could be for the interest of readers from the field of liver diseases, especially those focused in autoimmune liver diseases.
There are however minor issues that could be easily addressed by the authors.
· If possible, please, discuss in your opinion, why the prevalence of this disease is raising world-wide in these days.
· Please add a paragraph discussing what remains unknown for autoimmune hepatitis and give your perspectives for future studies or directions for the research of this disease.
· In the topic of fibrosis assessment, it would be desired to determine what parameters are stablished in the clinic and what are still in the research phase.
· Perhaps it is out of the scope of the paper, but some notions of available animal models of this disease would be desired for basic researchers.
· In figure 1, it seems that apoptotic bodies are a source of ROS. However, according to the text, “ROS are generated by myofibroblasts and Kupffer cells when they incorporate apoptotic bodies. ROS in turn stimulate the intrinsic apoptotic pathway and activate HSCs implicated in fibrogenesis”. Perhaps, in order to make the figure clearer, the text of “ROS” placed on the left of the apoptotic bodies should be deleted.
· Page 3 line 91. The abbreviation HSC have been previously described in page 2 line 66.
· Please add some references for the statements included in the paragraph placed in page 3 lines 95-99.
· Please add some references for the statements included in page 3 lines 102-103.
· Table 1. The specific procedures does not correspond to the general method. It seems that there has been some table formatting problems.
· Page 4 line 142. It is not clear for this referee the subject in the phrase “However, they have not yet been well detected in AIH”. Who are “they”?
· Please, define the abbreviations HBV and HCV for the first time (page 5 line 175).
· Page 6 line 231. The abbreviation US have been previously described in page 6 line 220.
· Please add a reference for the statement included in the paragraph placed in page 7 lines 291-293.
· Figure 2. Please add a reference if the figure has been obtained/modified
· Page 10 line 399, Page 10 line 416 and Page 11 line 438. The abbreviation AZA have been previously described in page 1 line 382.
· Page 11 line 453. The abbreviation “TNA-α” should be “TNF-α”.
· Please add a “full stop” at the end of page 11 line 479.
· Please, confirm in page 12 line 492 that the term “proprieties” is the one desired or it should be “properties”.

Author Response

(The authors gave the same response as above.)
